# The Accuracy of Winter Wheat Identification at Different Growth Stages Using Remote Sensing

Shengwei Liu [1,2], Dailiang Peng [2,3,*], Bing Zhang [2], Zhengchao Chen [2], Le Yu [4], Junjie Chen [1], Yuhao Pan [2,5], Shijun Zheng [2,5], Jinkang Hu [2,5], Zihang Lou [2,5], Yue Chen [6] and Songlin Yang [2,7]

1. School of Surveying and Land Information Engineering, Henan Polytechnic University, Jiaozuo 454000, China; 212004010029@home.hpu.edu.cn (S.L.); chenjj@hpu.edu.cn (J.C.)
2. Key Laboratory of Digital Earth Science, Aerospace Information Research Institute, Chinese Academy of Sciences, Beijing 100094, China; zb@radi.ac.cn (B.Z.); chenzc@radi.ac.cn (Z.C.); panyuhao20@mails.ucas.ac.cn (Y.P.); zhengshijun19@mails.ucas.ac.cn (S.Z.); hujinkang21@mails.ucas.ac.cn (J.H.); louzihang21@mails.ucas.ac.cn (Z.L.); yangsonglin211@mails.ucas.ac.cn (S.Y.)
3. International Research Center of Big Data for Sustainable Development Goals, Beijing 100094, China
4. Department of Earth System Science, Tsinghua University, Beijing 100084, China; leyu@tsinghua.edu.cn
5. College of Resource and Environment, University of Chinese Academy of Sciences, Beijing 100049, China
6. School of Electronic and Information Engineering, National Engineering Research Center for Agro-Ecological Big Data Analysis & Application, Anhui University, Hefei 230093, China; p20301132@stu.ahu.edu.cn
7. School of Electronic, Electrical and Communication Engineering, University of Chinese Academy of Sciences, Beijing 100049, China
* Correspondence: pengdl@aircas.ac.cn

**Abstract:** The aim of this study was to explore the differences in the accuracy of winter wheat identification using remote sensing data at different growth stages using the same methods. Part of northern Henan Province, China was taken as the study area, and the winter wheat growth cycle was divided into five periods (seeding-tillering, overwintering, reviving, jointing-heading, and flowering-maturing) based on monitoring data obtained from agrometeorological stations. With the help of the Google Earth Engine (GEE) platform, the separability between winter wheat and other land cover types was analyzed and compared using the Jeffries-Matusita (J-M) distance method. Spectral features, vegetation index, water index, building index, texture features, and terrain features were generated from Sentinel-2 remote sensing images at different growth periods, and then were used to establish a random forest classification and extraction model. A deep U-Net semantic segmentation model based on the red, green, blue, and near-infrared bands of Sentinel-2 imagery was also established. By combining models with field data, the identification of winter wheat was carried out and the difference between the accuracy of the identification in the five growth periods was analyzed. The experimental results show that, using the random forest classification method, the best separability between winter wheat and the other land cover types was achieved during the jointing-heading period: the overall identification accuracy for the winter wheat was then highest at 96.90% and the kappa coefficient was 0.96. Using the deep-learning classification method, it was also found that the semantic segmentation accuracy of winter wheat and the model performance were best during the jointing-heading period: a precision, recall, F1 score, accuracy, and IoU of 0.94, 0.93, 0.93, and 0.88, respectively, were achieved for this period. Based on municipal statistical data for winter wheat, the accuracy of the extraction of the winter wheat area using the two methods was 96.72% and 88.44%, respectively. Both methods show that the jointing-heading period is the best period for identifying winter wheat using remote sensing and that the identification made during this period is reliable. The results of this study provide a scientific basis for accurately obtaining the area planted with winter wheat and for further studies into winter wheat growth monitoring and yield estimation.

**Keywords:** winter wheat identification; remote sensing; random forest; deep learning; semantic segmentation; jointing-heading period

## 1. Introduction

As well as being the most widely distributed food crop in the world, with more than 40% of the world's population relying on it as their staple food, wheat also plays an important role in food production and supply, providing about 20% of overall human energy consumption [1,2]. China is the largest wheat producer and consumer in the world [3]. The area sown with winter wheat accounts for 20.8% of China's grain crop area, and the output of winter wheat accounts for 20.3% of China's total crop output [4]. However, China still faces many challenges in meeting its population's growing demand for food. Therefore, obtaining the area planted with winter wheat and its spatial distribution timely and accurately is an important part of agricultural monitoring and is very important to estimates of the winter wheat yield, ensuring food security, adjusting the structure of agriculture, the formulation of agricultural policies, and the promotion of agricultural production [5–9].

The traditional method of obtaining details of the area planted with winter wheat depended mainly on statistical reports. This was not only time-consuming and laborious, but also easily influenced by subjective human factors, which could lead to large discrepancies in the results. In recent decades, with the rapid development of satellite remote sensing technology, new opportunities for monitoring winter wheat have become available. Due to satellite data's wide coverage, the short revisit period, and low cost, the use of these data has become the main method of monitoring and obtaining the area planted with winter wheat [10,11]. It is of great scientific significance and practical value to obtain the spatial distribution and planting area of winter wheat timely and accurately at different scales using remote sensing technology [4]. Single-phase medium-to-low resolution remote sensing images are usually used for identifying winter wheat. Although this is a relatively simple and efficient way of performing the identification, the results are easily affected by the weather and the image resolution, which often causes the accuracy of the identification and extraction of the winter wheat to be low. As a result, many researchers began to use multi-temporal medium-to-low resolution remote sensing images to extract the area planted with winter wheat. Later, it was found that the differences between the growth cycles and phenology of different crops could be reflected in time-series of remote sensing data; as a result, vegetation index time-series data have been widely used in extracting the area of winter wheat. For example, many researchers have established models for extracting the winter wheat area based on MODIS NDVI and MODIS EVI time-series curves and achieved a high extraction accuracy [12–15]. Because of their high spatial resolution and wide coverage, the use of medium-to-low resolution images allows the construction of NDVI time-series covering large regions, thus enabling the extraction of the winter wheat area over large areas [16,17]. However, the constructed time series generally cover the complete growth period of winter wheat from sowing to harvesting, which means that the winter wheat area often cannot be extracted in a timely way when needed. In order to solve these problems, Wang et al. [18] and Guo et al. [19] used Landsat and MODIS data to replace the original NDVI time series with an incremental NDVI reconstruction algorithm that extracted as much useful information from the NDVI time series as possible and shortened the period over which the time series extended. In this way, good extraction results for winter wheat were achieved. Generally, medium-to-low resolution images are used for the identification of winter wheat over large areas, but it has been difficult to meet the increasingly refined requirements for the identification of winter wheat and the monitoring of the planting area. With the successive launches of high spatial resolution Earth observation satellites, such as the European Space Agency's Copernicus Sentinel-2A/B satellites, high-resolution images provide new opportunities for the more accurate extraction of the winter wheat area and the mapping of winter wheat crops [20]. In addition, high spatial resolution images clearly show crops' spatial structure, texture features, and edges [21] and have been used by many studies to identify winter wheat [22]. These studies still had some shortcomings: (1) They were limited to municipal regions and small areas

because high-resolution images are limited by their coverage and easily affected by clouds, and it is difficult to obtain high-quality images that cover the whole growth cycle of winter wheat over large areas. (2) They are based on remote sensing images acquired on specific dates and few attempts have been made to explore the accuracy of the identification of winter wheat in different growth periods [23]. (3) Although many algorithms have been applied to winter wheat identification, few methods discuss the differences of winter wheat identification at different growth periods. Given the importance of the monitoring of winter wheat to food security, it is crucial that the identification of winter wheat can be carried out effectively and accurately using images from the key growth periods of winter wheat. Therefore, how to find the best period on which to base the identification of winter wheat is particularly important. In one recent study, the separability between winter wheat and other land cover types at different growth periods in northern and central Anhui Province was calculated using the J-M method, and it was determined that the heading period was the best period on which to base the extraction of the winter wheat area [24]. In another study, time-aggregation techniques were used to combine Landsat-8 OLI and Sentinel-2 images to explore the identification of winter wheat during different growth periods in Shandong Province. It was determined that the use of data from the maturing and heading periods gave the best results [25]. However, in these studies, only a small number of features were used for the identification and classification, and the influence of other features such as the terrain and texture on the identification and classification of winter wheat was ignored. In addition, only one classification method, random forest classification, was used and no evaluation of the use of other methods, such as deep learning, for the identification of winter wheat in different growth periods was made.

The Google Earth Engine (GEE) is cloud-based global-scale geospatial analysis platform that allows a large number of remote sensing images to be processed quickly [26]. The GEE not only stores a large amount of publicly available remote sensing imagery, but also possesses very powerful cloud computing capability. Users can easily extract, call, and analyze remote sensing big data resources as well as carry out online calculations and processing. This provides new opportunities for the rapid classification of remote sensing imagery, crop extraction, area monitoring, etc. [27].

Therefore, the objective of this study was to explore the accuracy of winter wheat identification at different growth periods using remote sensing data. The specific goals were to (1) establish a random forest classification and extraction model for finding the best period on which to base the identification of winter wheat from Sentinel-2 images; (2) construct a deep learning U-Net semantic segmentation model for verifying the results of random forest; (3) calculate the area of winter wheat based on the best identification period, separately; (4) identify the advantages and uncertainties of winter wheat identification in this study.

## 2. Materials

### 2.1. Study Area

Anyang City, Puyang City, Xinxiang City, and Hebi City are located in the northern part of Henan Province (as shown in Figure 1); together, these constituted the study area used in this research. This area is mainly a plain that has a high elevation in the west and a low elevation in the east; the total area is about 22,032 km2. The study area has a warm temperate continental monsoon climate with four distinct seasons. The monthly average temperature in 2020 is between 8.9 and 15.6 °C, and the annual cumulative precipitation in 2020 is between 607.5 and 847.7 mm. The rainy season covers June to August [28]. Large parts of the area are irrigated and the winter wheat yield is high, meaning that the area constitutes one of the centers of high-quality winter wheat production in China. Besides winter wheat, the other main crops grown in the area include corn, garlic, and vegetables.

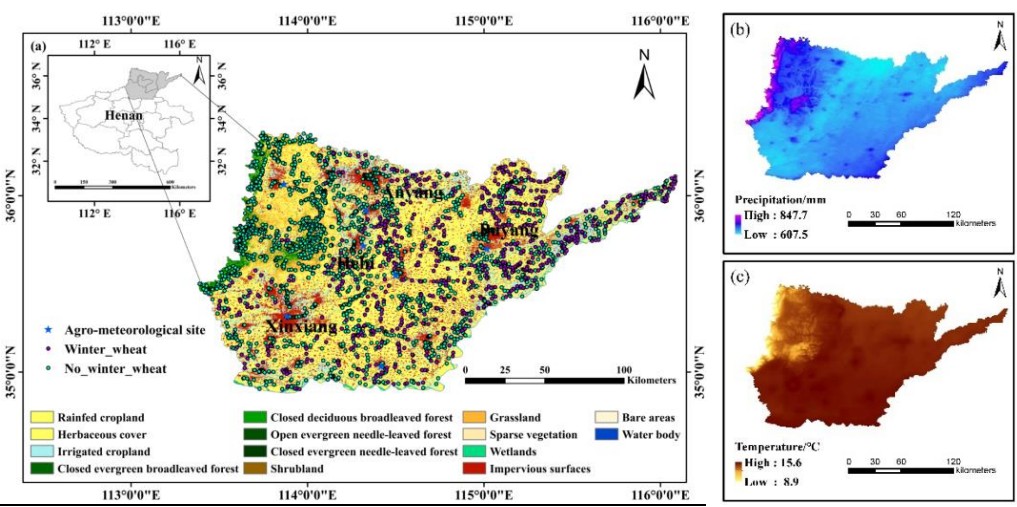

**Figure 1.** Geographical location of the study area: (**a**) Distribution of sample points. The main map is based on the 30-m global land cover product fine classification system for 2020, (**b**) Annual cumulative precipitation data in 2020, (**c**) Monthly average temperature data in 2020.

### *2.2. Datasets*

#### 2.2.1. Remote Sensing and Terrain Data

In this study, Sentinel 2 level-2A surface reflectance product (MSI) data acquired between 10 October 2019 and 10 June 2020 were used. These data had undergone radiometric correction, atmospheric correction and orthorectification. The reflected spectral features were constructed by calling 10 bands of the product in GEE. These 10 bands included the red, green, blue, and near-infrared bands (band 4, band 3, band 2, and band 8), which have a spatial resolution of 10 m, together with the red edge 1, red edge 2, red edge 3, narrow-near infrared, short-wave infrared 1, and short-wave infrared 2 bands, which have a 20-m spatial resolution (band 5, band 6, band 7, band 8A, band 11, and band 12).

Digital elevation data that were acquired by the Shuttle Radar Topography Mission allowed the production of digital elevation models with near-global coverage [29]. The SRTMGL1_003 product is provided by the NASA JPL at a resolution of 1 arcsecond (approximately 30 m). In this study, this product was used in GEE to construct the terrain features of the study area, including the altitude, slope, aspect, and mountain shadow.

#### 2.2.2. Agrometeorological Site Data

The agrometeorological site data used in this study included details of the winter wheat development periods. Based on the data acquired from seven agrometeorological sites in the study area, the dates corresponding to the five different growth periods for 2019 to 2020 are as shown in Table 1. The locations of the agrometeorological sites are shown in Figure 1.

**Table 1.** Dates corresponding to the five different growth periods in the study area.

| Growth Period | Dates |
| --- | --- |
| Seeding-tillering | 10 October 2019–13 December 2019 |
| Overwintering | 1 January 2020–20 February 2020 |
| Reviving | 21 February 2020–24 March 2020 |
| Jointing-heading | 25 March 2020–24 April 2020 |
| Flowering-maturing | 25 April 2020–10 June 2020 |

#### 2.2.3. Sample Data

Winter wheat and non-winter wheat sample points were also collected for use in this study. These data were randomly collected within the study area. The non-winter wheat samples mainly included garlic, vegetables, and corn. In addition to field collection, visual

interpretation was also used. That is, points were marked using the GEE; the selected points mainly included ones with obvious characteristics corresponding to forests, water, buildings, and roads. Data from all of the sample points were converted to KML format and imported into Google Earth for inspection. After removing obvious errors, 805 samples of winter wheat and 2329 samples of non-winter wheat remained; the spatial distribution of these samples is shown in Figure 1. Based on the previous study, the sample points were then randomly divided into two parts: 70% were used for training and classification and 30% for accuracy evaluation [30], as shown in Table 2.

**Table 2.** Number and proportion of samples used for training and validation.

| Class | No. of Training Samples | No. of Validation Samples | Proportion |
|---|---|---|---|
| Winter wheat | 568 | 237 | |
| Buildings and roads | 461 | 196 | |
| Forest | 426 | 185 | 7:3 |
| Other vegetation | 389 | 166 | |
| Water | 354 | 152 | |

## 3. Methods

### 3.1. Application of the Random Forest Classification Method

3.1.1. Data Preprocessing

Using the GEE platform, the Sentinel-2 remote sensing images corresponding to the five different growth periods were obtained by calling the appropriate image collection (COPERNICUS/S2_SR). Cloud removal was realized by using the quality control band (QA60) mark; the spectral bands with spatial resolutions of 10 m and 20 m that were described above were then selected. Finally, the ten bands were stacked and cropped over the area of interest.

3.1.2. Use of the J-M Distance to Calculate the Separability between Different Land Cover Types

The J-M distance is a spectral separability index based on conditional probability theory and is usually used to measure the separability between classes. Previous studies indicated that the J-M distance can more appropriately describe the differences between classes than other indexes and that it is an effective measure for evaluating the separability between different land-cover training samples [31,32]. In this study, the J-M distances between five land cover types were calculated to compare and analyze the separability between winter wheat and other land cover types during different growth periods. The five land cover types included winter wheat, buildings and roads, other vegetation (garlic, corn, vegetable fields, grass), forest and water. For each cover type, 10% of the dataset was selected as the sample using equal proportion random sampling to calculate the J-M distance. For features that conform to a normal distribution (features that did not conform to a normal distribution were considered to have poor separation and classification was not attempted), the J-M distance can be calculated as [33]:

$$J = 2(1 - e^{-B}),\qquad(1)$$

$$B = \frac{1}{8}(m_i - m_j)^T \left(\frac{\sum i + \sum j}{2}\right)^{-1}(m_i - m_j) + \frac{1}{2}In\left[\frac{\frac{\sum i + \sum j}{2}}{(|\sum i||\sum j|)^{\frac{1}{2}}}\right]\qquad(2)$$

where $B$ represents the Bhattacharyya distance, $m_i$ and $m_j$ represent the mean values of the spectral reflectance of classes $i$ and $j$, respectively. $\sum i$ and $\sum j$ are the unbiased estimates for the covariance matrices of $i$ and $j$, respectively. The J-M distance has a range of 0.0 to 2.0, with 0.0 meaning that the two categories are almost inseparable and 2.0 meaning that they

can be completely separated. The higher the value of *J*, the better the separability between the cover types.

### 3.1.3. Feature Construction

Ten original bands from Sentinel-2 remote sensing images were selected for the construction of the spectral features; the commonly used indexes features were obtained from the band math. The index features consisted of the Normalized Difference Vegetation Index (NDVI) [34], Enhanced Vegetation Index (EVI) [35], Soil Adjusted Vegetation Index (SAVI) [36], Normalized Difference Water Index (NDWI) [37], Modified Normalized Difference Water Index (MNDWI) [38], and Normalized Difference Building Index (NDBI) [39]. The Sentinel-2 red-edge bands are important in agricultural applications [40], and the position of the red edge is an important index that can be used to measure the chlorophyll content of leaves. Using the reflectance of the red-edge region to calculate a vegetation index can thus improve the classification accuracy [41]. Therefore, a red edge normalization index (RENDVI) and a red edge position index (REP) were also constructed by using the red-edge bands of Sentinel-2 data [42]. The formulae for calculating the different indexes are listed in Table 3.

**Table 3.** Description of spectral features.

| Name | Expression |
|---|---|
| Normalized Difference Vegetation Index (NDVI) | $\frac{B8-B4}{B8+B4}$ |
| Enhanced Vegetation Index (EVI) | $2.5 * \frac{B8-B4}{B8+6*B4-7.5*B2+1}$ |
| Soil Adjusted Vegetation Index (SAVI) | $1.5 * \frac{B8-B4}{B8+B4+0.5}$ |
| Normalized Difference Water Index (NDWI) | $\frac{B3-B8}{B3+B8}$ |
| Modified Normalized Difference Water Index (MNDWI) | $\frac{B3-B11}{B3+B11}$ |
| Normalized Difference Building Index (NDBI) | $\frac{B11-B8}{B11+B8}$ |
| Red Edge Normalized Difference Vegetation Index (RENDVI) | $\frac{B8-B6}{B8+B6}$ |
| Red Edge Position (REP) | $705 + 35 * \frac{0.5*(B4+B7)-B5}{B6-B5}$ |

Where B2, B3, B4, B5, B6, B7, B8, and B11 represent the reflectance values of the blue, green, red, red edge 1, red edge 2, red edge 3, and near-infrared bands and short-wave infrared band 1, respectively.

The vegetation indexes, building index, and water body index were added to each image as independent bands; all of the images from this growth period were then medianly synthesized to obtain a synthesized image with 18 bands. Because there are many mountains in the west of the study area, there are few winter wheat planting areas here and they are difficult to identify. In order to improve the classification accuracy of the winter wheat planting areas, terrain features were added to the training features. In addition, winter wheat has a continuous, regular texture in remote sensing images. In order to better extract the spatial distribution of winter wheat in the study area, texture features were also used. Four terrain features—altitude, slope, aspect and mountain shadow—were constructed by calling SRTMGL_003 data in GEE. Image texture is produced by the repeated appearance of pixel gray levels in spatial position. Gray level co-occurrence matrix (GLCM) is a common method to describe the texture by studying the spatial correlation characteristics of the gray levels [43]. Because the near-infrared band plays an important role in the remote sensing of vegetation, and as the reflection from vegetation in the near-infrared region is extremely obvious due to the effects produced by the internal structure of leaves, the near-infrared band (B8) of the Sentinel-2 data was used to calculate the texture features [44]. Calling the GLCM texture feature function in GEE allowed six texture features to be quickly calculated. These features included the angle second moment, contrast, correlation, variance, inverse

difference moment, and entropy. All 28 features were then integrated into one image and resampled to a 10-m spatial resolution.

### 3.1.4. Random Forest Algorithm and Accuracy Evaluation Index

The random forest is a classifier that uses multiple decision trees. It adopts the bootstrap step-by-step sampling strategy to randomly extract two-thirds of the data from the original data set to create training sets, and decision trees are established for each training set. In each sampling process, one-third of the data is not extracted, and this part of the data is used for the unbiased estimation of internal errors, thus generating an OOB error that can be used to evaluate the classification accuracy of the random forest [45,46]. The random forest algorithm is widely used in the classification of winter wheat in different types of satellite data because of its high efficiency, lack of sensitivity to noise, strong adaptability, and ability to evaluate the importance of each feature in the classification. In addition, compared with other classification methods, the random forest classifier can process higher-dimensional data and achieve a higher accuracy [2]. Therefore, in this study, the ee.Classifier.smileRandomforest function was used to build a random forest classification model. Some previous studies have shown that different parameterization schemes of random forest model have a limited influence on classification accuracy [47]. To reduce calculation amount and acquire relatively higher classification accuracy [48,49], the number of decision trees was set to 200; all the other parameters were set to their default values.

The use of the confusion matrix is a standard method for evaluating the accuracy of remote sensing image classification results [50]. In this paper, the errorMatrix function was used in GEE to calculate the confusion matrix. Four evaluation indexes, the user accuracy (UA), producer accuracy (PA), overall accuracy (OA) and kappa coefficient, were selected to evaluate the accuracy of the identification of winter wheat based on the random forest classifier.

### 3.2. Application of the Deep Learning Classification Method

### 3.2.1. Training, Validation and Test Data Sets

Due to the lack of labeling data for winter wheat in the study area, the U-Net training process required image datasets and their corresponding labels. Therefore, particular rectangular areas from within the study area were selected to construct the image datasets that applied for the training, validation, and testing. The selected areas ① contained obvious differences in ground features, ② contained all of the land cover types described earlier, and ③ included each city in the study area. Following this selection, using ArcGIS, the Sentinel-2 median composite images for the whole growth stages were cropped by rectangular polygon features of the selected areas. The winter wheat labels for these selected areas were created by referring to the composite images for the five growth stages and the field data points; the id field values of the labels were then modified and the spatial references of the labels were set. The winter wheat label vector files were converted to raster files and point features randomly created in the rectangular areas. Graphic buffers were then created for the point features to generate square vector data with dimensions of 1280 m × 1280 m; the boundaries of these data were set to lie within the winter wheat label. Then, a custom-built model and the square vector data were used to cut the Sentinel-2 median composite images (consisting of the red, green, blue and near-infrared bands) corresponding to the five growth periods and the previously generated raster labels into batches. Finally, the cropped remote sensing images and labels were adjusted to a size of 128 pixels × 128 pixels to produce the finished data set. The data set was randomly divided into a training set, validation set and test set using the ratio 7:2:1. This gave 3424 samples in the training set, 857 samples in the validation set and 458 samples in the test set. Before being used as the input to the training algorithm, the training set images were normalized and subjected to data augmentation; that is, new images were generated by adjusting the color and by rotating and symmetrizing the images [51].

The Sentinel-2 median composite images for the five growth stages were cut into $512 \times 512$ pixels pieces, and the sub-images whose values were all background values were removed. This produced the final data sets to be classified for each growth stage.

### 3.2.2. U-Net Network Parameter Setting and Accuracy Assessment

The structure of the U-Net network includes down-sampling and up-sampling, and the complete network has the appearance of a letter 'U'. In the first part, the features of the input image are extracted by the convolution layers and maximum pooling layers; each $3 \times 3$ convolution layer is followed by a ReLU activation function and a $2 \times 2$ maximum pooling operation. In the second part, after deconvolution, the result is spliced with the corresponding feature map to restore the resolution. In the final output layer, the 1*1 convolution kernel is used [52,53].

The hardware environment used for the training, validation and testing consisted of an Inter (R) Xeon (R) Gold 6226R 2.9 GHz 16-core processor, an NVIDIA GeForce RTX 3090 24 GB graphics card and a 256 GB Hynix DDR4 memory. The software environment consisted of python 3.7 and pytorch 1.7.1.

The learning rate is often a difficult parameter to choose the setting for, but it is also one of the most important parameters. In this experiment, the learning rate was set to $1 \times 10^{-7}$, and the learning rate was adaptively adjusted by calling ReduceLROnPlateau in pytorch; that is, when the loss function of the validation set no longer decreased after 20 epochs, the learning rate was adjusted to be one tenth of the original. The number of classes was set to 2, the batch size to 128, the number of bands to 4, the number of training epochs to 400 and the size of the input image to $128 \times 128$ pixels. The Adam algorithm was selected on the network optimizer and the cross-entropy loss function was chosen as the loss function. In addition, in order to prevent over-fitting by the model and gradient disappearance, L2 regularization with a coefficient of $5 \times 10^{-4}$ was included in the training process.

In order to verify the accuracy of the U-Net model's semantic segmentation, the precision, recall, *F1-score*, *IoU* and accuracy were used to quantitatively evaluate the results for the semantic segmentation of winter wheat using the five test sets corresponding to different growth stages [54]:

$$IoU = \frac{TP}{TP + FP + FN}, \tag{3}$$

$$Accuracy = \frac{TP + TN}{TP + TN + FP + FN}, \tag{4}$$

$$Precision = \frac{TP}{TP + FP}, \tag{5}$$

$$Recall = \frac{TP}{TP + FP} \text{ and} \tag{6}$$

$$F1 - Score = 2 * \frac{precision * recall}{precision + recall}. \tag{7}$$

Here, *TP* represents the number of true positives, *TN* the number of true negatives, *FP* the number of false positives and FN the number of false negatives. The *F1 score* is the harmonic average of precision and recall.

### 3.3. Extraction of the Winter Wheat Planting Area and Accuracy Verification

The total area of winter wheat in the study area was calculated by adding the area of winter wheat in each pixel [55]. The accuracy of the results for the area sown with winter wheat was calculated as the ratio of the estimated area to the ground-truth value, which was taken from the statistical yearbook of Henan Province for 2020. The accuracy was calculated as [56]

$$P = 1 - \frac{|S - S'|}{S'} * 100\%, \tag{8}$$

where *P* represents the accuracy of the extracted area, *S* represents the area planted with winter as extracted by the two methods used in this study and *S'* represents the real area planted with winter wheat.

## 4. Results and Analysis

### 4.1. Analysis of the Random Forest Classification Results

#### 4.1.1. Analysis of the J-M Distance Results

In this study, the J-M distance was selected in GEE to calculate the separability between winter wheat and the other land-cover types during different growth periods. The results of these calculations are shown in Table 4. It can be seen that the separation between winter wheat and forest and water is good in the study area; however, the separability between winter wheat and buildings and roads and between winter wheat and other vegetation (corn, garlic, vegetable fields and grassland) is poor. The separability between winter wheat and other vegetation is highest during the jointing-heading period (1.98) but is only 1.46 during the seeding-tillering period. These results show that the jointing-heading period is the optimum period for distinguishing winter wheat from the other land-cover types.

**Table 4.** Jeffries-Matusita distance between winter wheat and other land-cover types.

| Training Sample Class | Seeding-Tillering | Overwintering | Reviving | Jointing-Heading | Flowering-Maturing |
|---|---|---|---|---|---|
| Buildings and roads | 1.89 | 1.93 | 1.97 | 1.99 | 1.95 |
| Forest | 1.96 | 1.99 | 1.99 | 1.99 | 1.99 |
| Other vegetation | 1.46 | 1.73 | 1.85 | 1.98 | 1.89 |
| Water | 1.99 | 1.99 | 1.99 | 1.99 | 1.99 |

#### 4.1.2. Analysis of the Accuracy Evaluation Results

The results of the evaluation of the random forest classification accuracy are shown in Figure 2. From the figure, it can be seen that the accuracy of the identification of winter wheat during the five growth periods is relatively high: the overall accuracy is above 91% and the kappa coefficient is above 0.89, which shows that the overall accuracy of the random forest classification is high and the model performance is good. The values of the overall accuracy and kappa coefficient (96.69% and 96.10%, respectively) are higher during the jointing-heading period than during the other four periods, indicating that identification of winter wheat is better during this period. In contrast, the values of the overall accuracy and kappa coefficient (91.99% and 89.90%, respectively) during the seeding-tillering period are lower than during the other four periods, which indicates that the error in the identification of winter wheat during this period is large. This is because winter wheat experiences vigorous growth during the jointing-heading period and thus contrasts strongly with other surface features and is easy to identify. During the seeding-tillering period, however, the winter wheat grows slowly and is not so easily distinguishable from other features, which leads to a low identification accuracy. It can also be seen from Figure 2 that, except for the seeding-tillering and overwintering periods, the user accuracy is relatively high, which indicates that the accuracy of the identification of winter wheat in the verification samples in these periods is good. In terms of the producer accuracy, the jointing-heading period has the highest value (95.36%) and the seeding-tillering period the lowest value (91.14%). This indicates that there is a certain amount of omission error during these five growth stages.

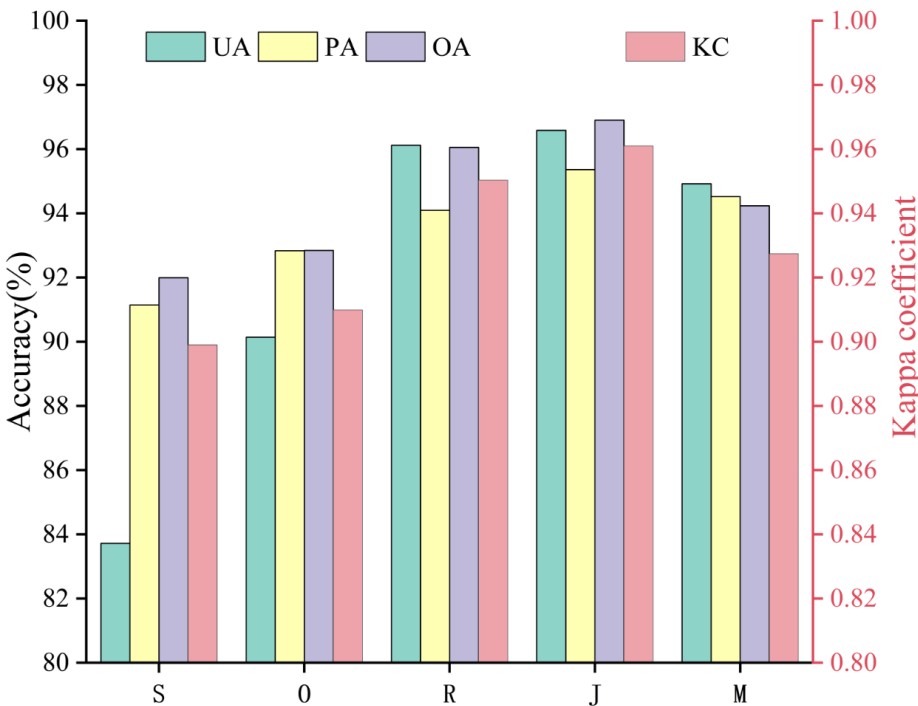

**Figure 2.** Results for the accuracy achieved using the random forest algorithm. S, O, R, J and M stand for the seeding-tillering, overwintering, reviving, joining-heading and flowering-maturing periods, respectively. The user accuracy (UA) (blue), producer accuracy (PA) (yellow) and overall accuracy (OA) (purple) are plotted on the left-hand vertical axis. The kappa coefficient (pink) is plotted on the right-hand vertical axis.

### 4.1.3. Winter Wheat Mapping Using Random Forest

Taking the map of the spatial distribution of winter wheat at the jointing-heading stage as the reference image (Figure 3a), it was subtracted from the winter wheat classification map corresponding to other periods to generate difference maps (Figure 3b–e) to display the differences in the areas identified as winter wheat during the different growth periods. In the difference maps, green represents TP (true positive; that is, identified as winter wheat in both periods), red represents FP (false positive; that is, identified as not being winter wheat during the jointing-heading period but identified as winter wheat during the other periods) and blue represents FN (false negative; that is, identified as winter wheat during the jointing-heading period but not as winter wheat during the other periods). From the green, red and blue areas in the difference map, it can be seen that there is little difference between the four other periods and the jointing-heading period; however, the sizes of the commission and omission errors vary. Further details are given in the bar charts, in which I represents the ratio of the number of correctly predicted winter wheat pixels to the union of the predicted number and the true number of winter wheat pixels (TP + FP + FN), II represents the ratio of the number of wrongly predicted winter wheat pixels to the union of the number of predicted winter wheat pixels and the true number, and III represents the ratio of the number of 'missing' winter wheat pixels to the union of the number of predicted winter wheat pixels and the true number. This means that, the higher I is, the smaller the difference between the amount of winter wheat predicted by the model in these two periods; the higher II is, the more serious the incorrect detection of winter wheat in these two periods; and the higher III is, the greater the amount of winter wheat that the model has failed to detect. As can be seen from Figure 3, compared with the jointing-heading period, the greatest difference in the number of pixels identified as winter wheat is for the seeding-tillering period: in this case, I is only 76.2%, and II and III are 10.1% and 13.7% respectively, this indicates serious commission and omission errors, the reviving period has the smallest difference: I is then 87.6% and the value of II is only 2.7%.

It can also be seen from the difference maps that the areas with the greatest differences are concentrated near the boundary between Anyang City and Puyang City as well as in the west of Xinxiang City and the east of Puyang City. Satellite imagery shows that these areas are mostly mountainous areas with complicated terrain and fragmented winter wheat planting areas.

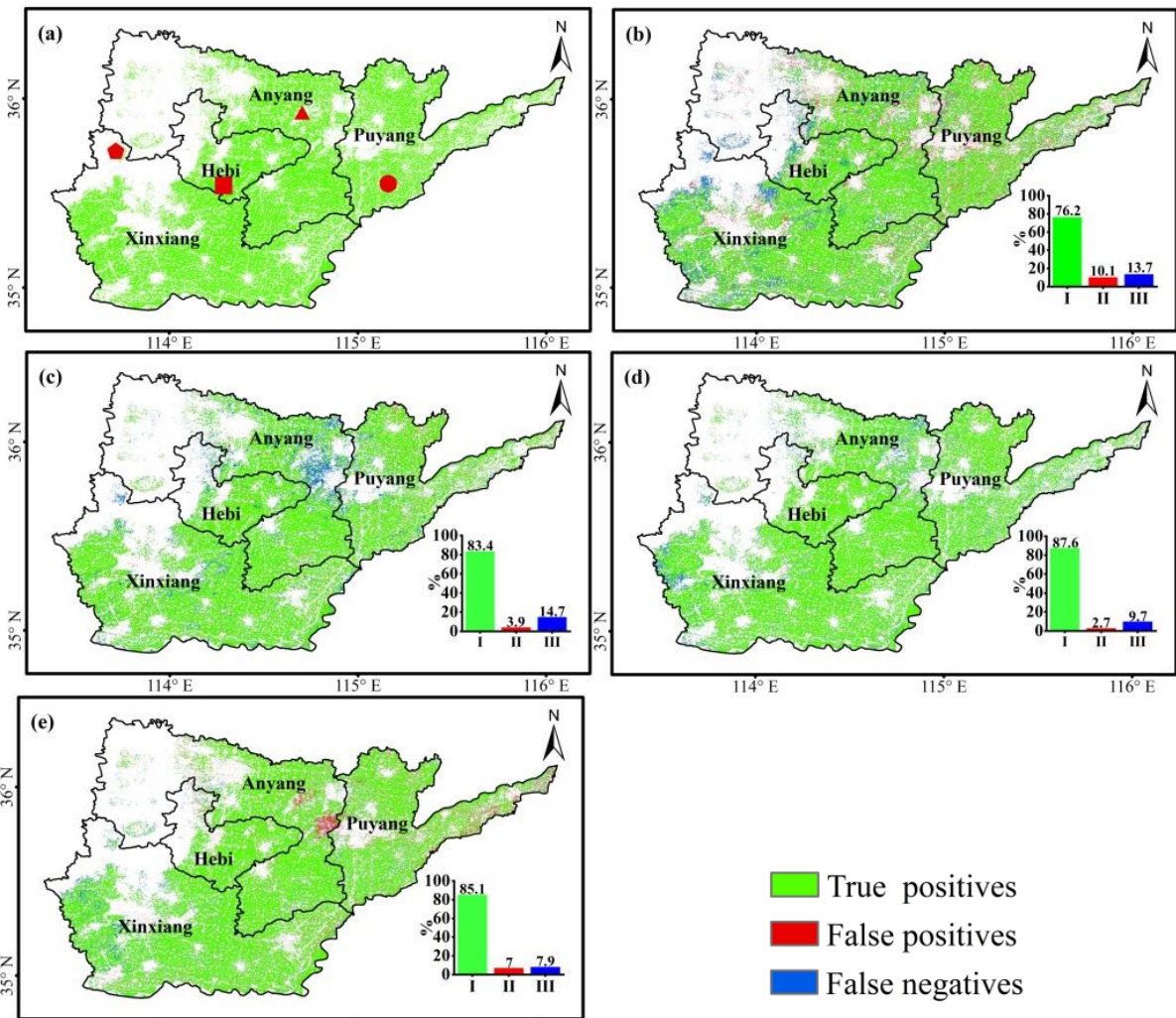

**Figure 3.** Winter wheat spatial distribution from five growth periods and their difference produced by the random forest classification. (**a**) The spatial distribution of winter wheat at the jointing-heading stage (reference image), it was subtracted from the winter wheat classification corresponding to other periods to generate the difference maps (**b**–**e**). In the bar charts, I, II and III represent the ratio of the number of correctly predicted, wrongly predicted and 'missing' winter wheat pixels to the union of the number of predicted winter wheat pixels and the true number, respectively. The red square, circle, triangle and pentagon represent the locations of the subsets shown in Figure 4.

In order to further explore the differences in the identification of winter wheat, four representative winter wheat planting areas (each with a size of 512 × 512 pixels) were selected for analysis (the locations of these subsets are shown in Figure 3a), and the random forest classification results for these areas are shown in Figure 4. Areas (a) and (b) represent densely planted areas of winter wheat and flat terrain, whereas (c) and (d) represent fragmented areas of winter wheat planted on complex terrain. From Figure 4, it is clear that the identification of winter wheat during the jointing-heading period is the best. In addition, in the areas with flat terrain and dense planting, the period has a good identification ability; however, in the fragmented areas of winter wheat and complex terrain, the commission

and omission errors during the other periods are more obvious than during the jointing and heading period, with the errors during the seeding-tillering period being the most serious. For example, in area c, an area of fragmented planting, the amount of incorrect detection of winter wheat is obvious, whereas in area d, a mountainous area with complex terrain, the missed detection of winter wheat is obvious. Therefore, the terrain is also an important factor that affects the accuracy of the detection of winter wheat based on remote sensing data and random forest classification. The more complex the terrain and the greater the degree of land-use fragmentation, the lower the accuracy of the detection.

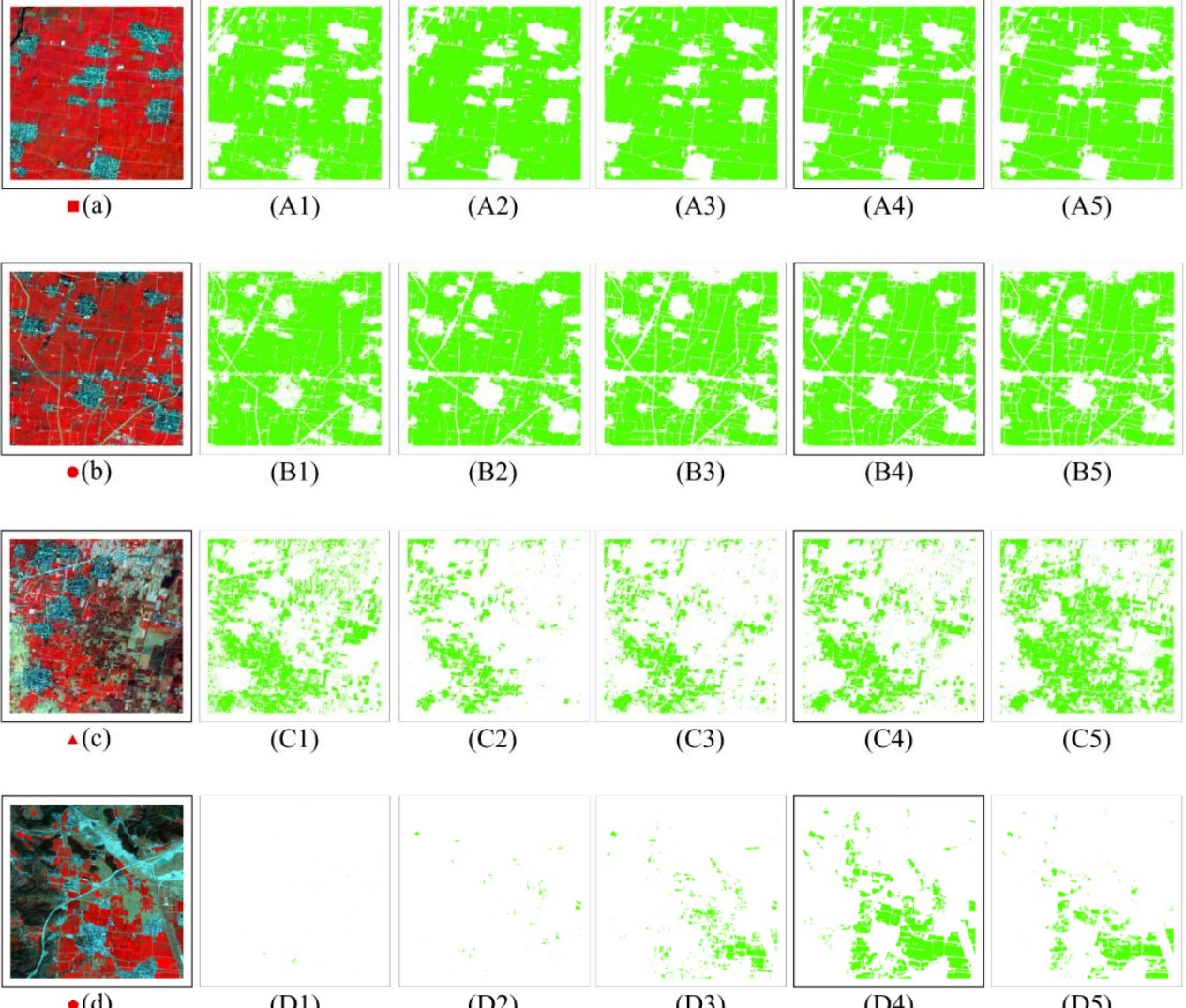

**Figure 4.** Results for the identification of winter wheat in four representative areas of the study area using random forest classification: (**a**,**b**) represent, respectively, winter wheat areas with dense planting, flat terrain; (**c**,**d**) represent, respectively, winter wheat areas with fragmented land use and complex terrain (Sentinel-2 false color composite images, bands 8/2/3). Parts (1) to (5) in each row show the pixels identified as winter wheat using the random forest classification during the seeding-tillering, overwintering, reviving and jointing-heading period, respectively.

### 4.2. Analysis of Deep Learning Classification Results

#### 4.2.1. Analysis of the Model Performance

Using the same U-net network model framework, the training and validation data sets for each growth period were trained separately. The loss function and accuracy of the training and validation sets of each growth period obtained during the training process are

shown in Figure 5. It can be seen from the figure that all five U-Net models reach stability after 400 training epochs. Near the beginning of the training, the loss function curve of training and validation set decreased obviously, and the accuracy of training and validation set also increased rapidly. By the time of training for 50 times, the decreasing speed of the loss function curves of both the training and validations quickly begin to flatten out until the accuracy no longer increases. During the process of the model training, the accuracy of the validation sets was set as the monitoring object. After the models had reached stability—that is, when the accuracy of the validation sets had reached a maximum—the models were saved under the optimal weights. This corresponded to epochs 398, 396, 390, 391 and 394 for parts (a), (b), (c), (d) and (e) in Figure 5, respectively.

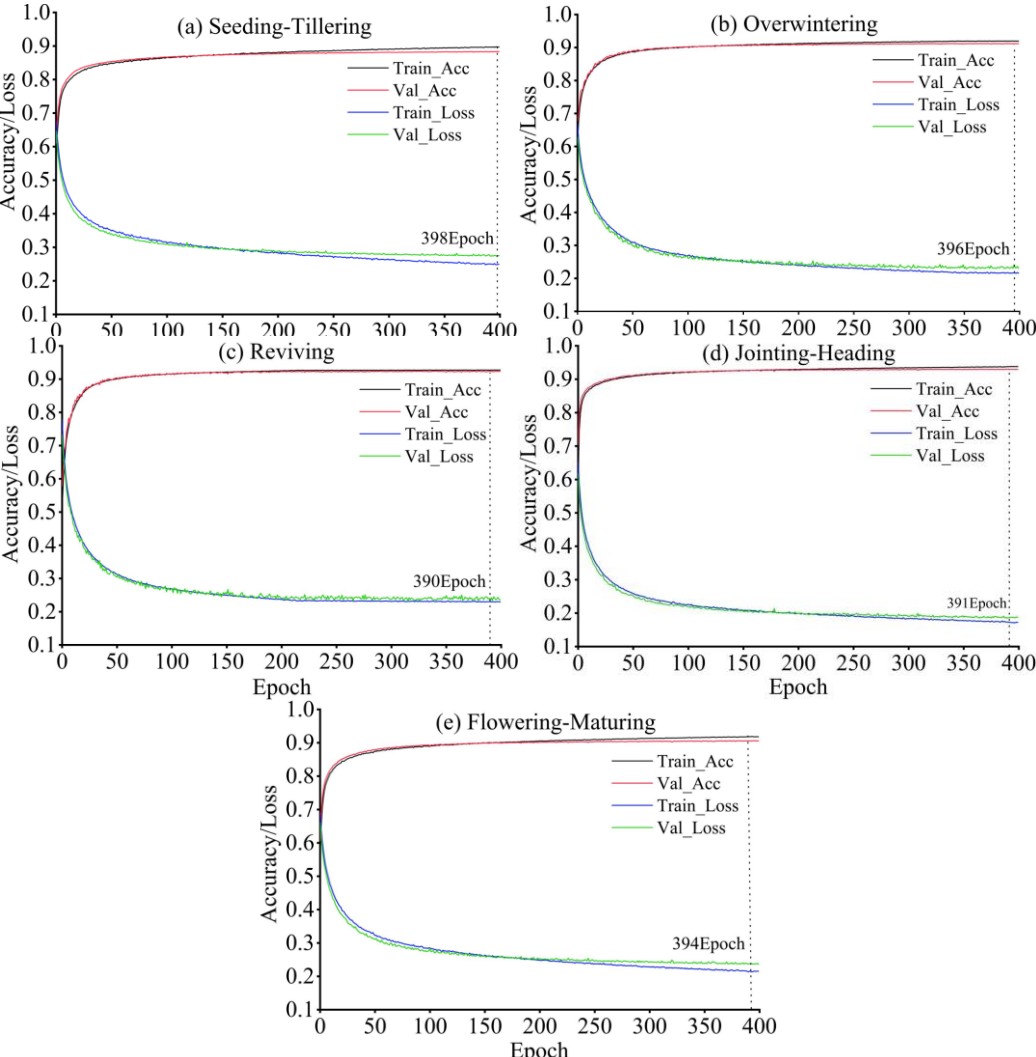

**Figure 5.** Accuracy and loss functions of the training and validation sets for different growth periods. The black curve represents the training set accuracy, the red curve represents the validation set accuracy, the blue curve represents the loss function of the training set and the green curve represents the loss function of the validation set. The dotted line marks the number of training epochs corresponding to the optimal weight.

The semantic segmentation of the test data sets corresponding to the five growth periods was carried out by the optimal trained models. The accuracy evaluation results for the test sets that were obtained are shown in Figure 6. The performance metrics of the jointing-heading period test set are the best overall: the IoU and F1-score for this test set are between 1.40% and 9.50% and between 0.80% and 5.70% higher than for the other four test sets, which shows that the model for the jointing-heading period produced the

best semantic segmentation results and the most accurate winter wheat identification. In contrast, the performance of the seeding-tillering period model is clearly worse than that of the other four models and this model has the lowest value for each evaluation metric. The IoU is below 0.8, which indicates that the classification of winter wheat produced by this model was poor.

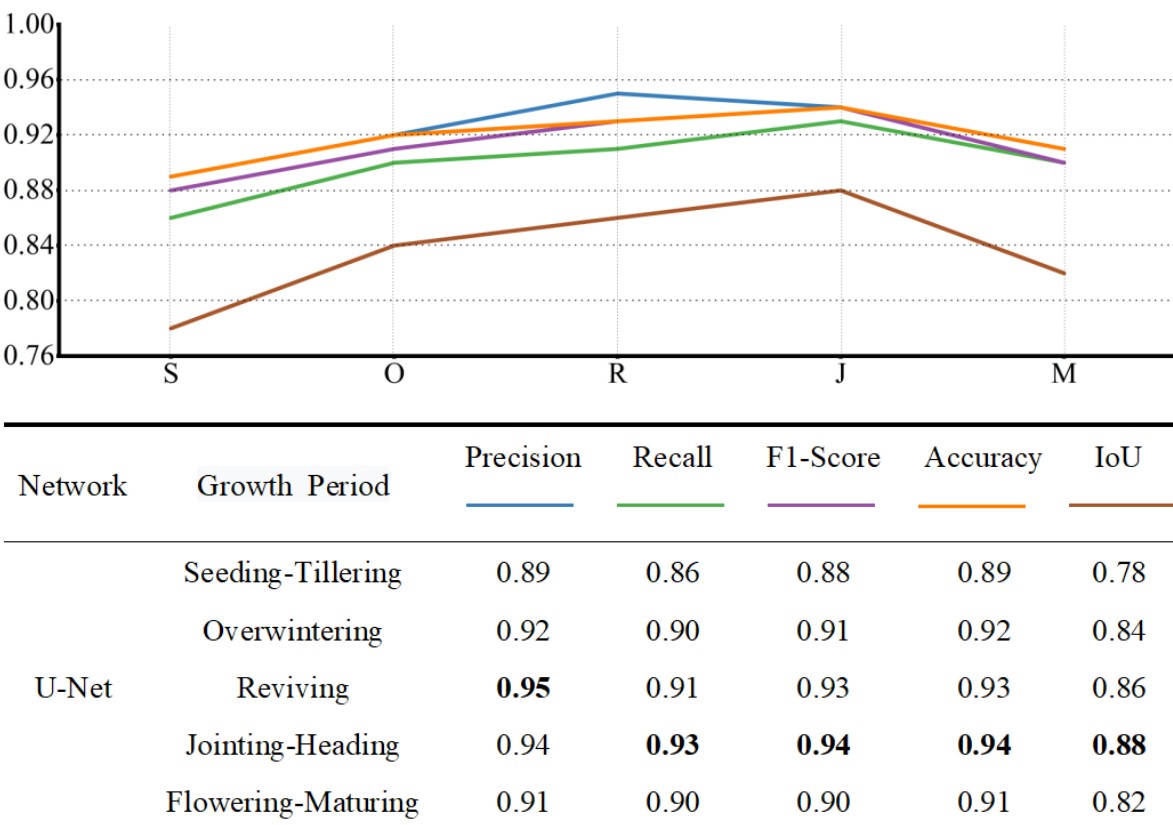

| Network | Growth Period | Precision | Recall | F1-Score | Accuracy | IoU |
|---------|---------------|-----------|--------|----------|----------|-----|
| U-Net | Seeding-Tillering | 0.89 | 0.86 | 0.88 | 0.89 | 0.78 |
| | Overwintering | 0.92 | 0.90 | 0.91 | 0.92 | 0.84 |
| | Reviving | **0.95** | 0.91 | 0.93 | 0.93 | 0.86 |
| | Jointing-Heading | 0.94 | **0.93** | **0.94** | **0.94** | **0.88** |
| | Flowering-Maturing | 0.91 | 0.90 | 0.90 | 0.91 | 0.82 |

**Figure 6.** Accuracy of the identification of winter wheat using the test set. S, O, R, J and M represent the seeding-tillering, overwintering, reviving, jointing-heading and flowering-maturing periods, respectively. The numbers in bold indicate the best value for each evaluation index.

As the number of images in the test set was large, it is impossible to display all of them. Therefore, four representative images were selected to analyze the accuracy of the identification of winter wheat using the deep learning method. The selected areas are shown in Figure 7. In the area shown in Figure 7a, the terrain is flat, the winter wheat grows well and is densely planted; in the area shown in Figure 7b, several land-cover types are present—besides winter wheat, there are weeds, buildings, roads, rivers and wasteland; in the area shown in Figure 7c, the winter wheat planting is sparse, and there are large areas of wasteland and a large number of houses; finally, in the area shown in Figure 7d, the terrain is complex, there is a small area planted with winter wheat, and most of the rest of the area consists of bare land and high mountains.

As can be seen from Figure 7, using the U-Net network, good semantic segmentation results were obtained for the different winter wheat planting areas in different scenes. For areas containing both flat terrain or mountains with complex terrain, the deep learning model identified areas of winter wheat well in all five growth periods. The ability to identify and classify winter wheat is the finest and the model performance is the best at jointing-heading period. The identification of winter wheat during the seeding-tillering period is the poorest.

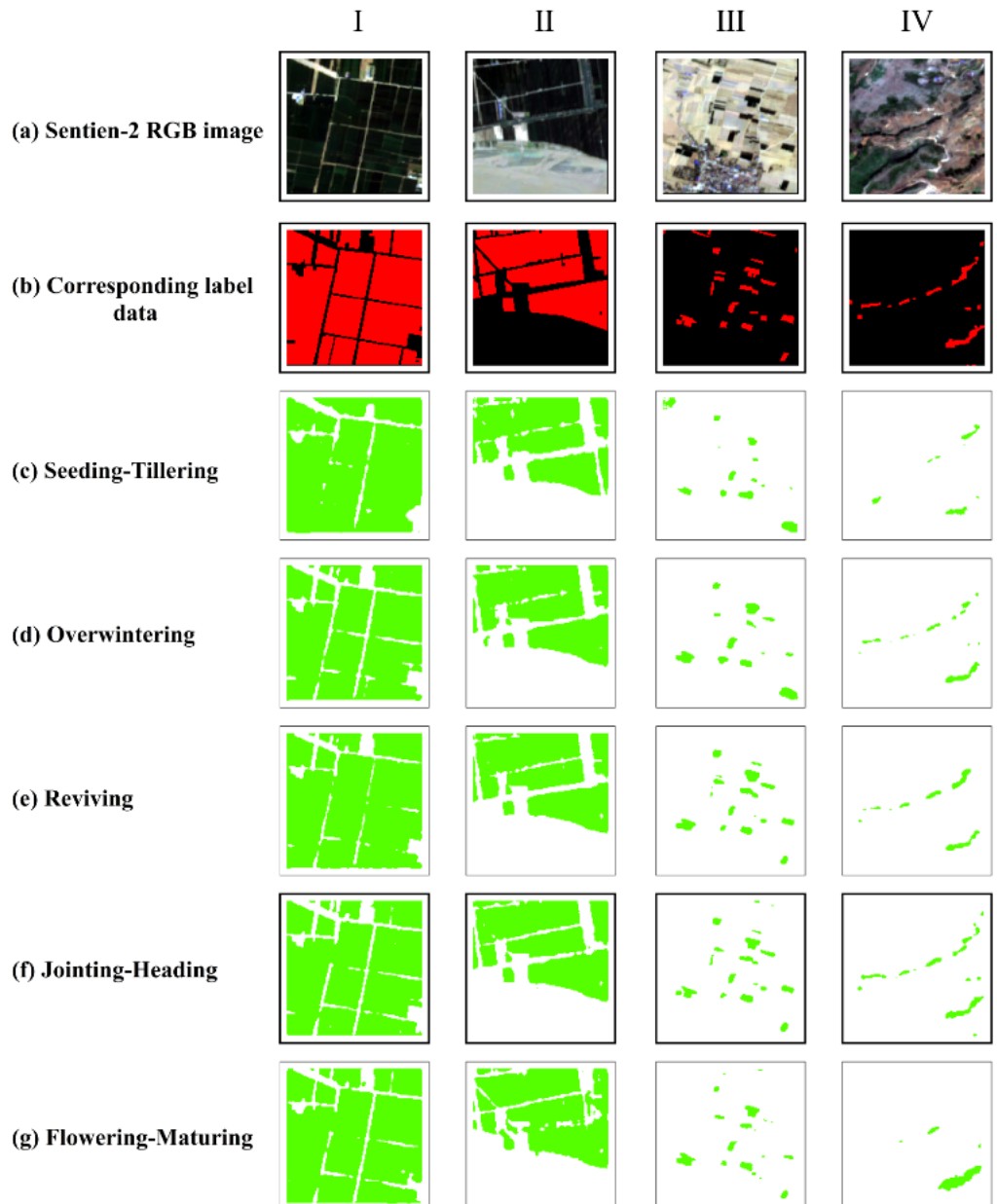

**Figure 7.** Results of the semantic segmentation of winter wheat using the test set. I-IV, respectively, represent winter wheat planting areas in an area of flat terrain, an area with multiple land-cover types, an area of fragmented terrain and an area of complex terrain. (**a**) True color Sentinel-2 composite images; (**b**) corresponding sample labels. (**c–g**), respectively, show the results of the semantic segmentation of the sample data for the seeding-tillering, overwintering, rejuvenation, jointing-heading and flowering-maturing periods.

### 4.2.2. Winter Wheat Mapping Using Deep Learning

The model parameters under the optimal weight for each growth period were called, and the Sentinel-2 data sets of the study area that were to be classified were trained. The training results were then spliced together for each growth period to generate the winter wheat classification map. Then, taking the winter wheat classification map for the jointing-heading period (Figure 8a) as the reference image, it was subtracted from the winter wheat classification map at the other periods to generate difference maps (Figure 8b–e). Further details are given in the bar charts, in which I represents the ratio of the number of correctly predicted winter wheat pixels to the union of the predicted number and the true number of winter wheat pixels (TP + FP + FN), II represents the ratio of the number of wrongly

predicted winter wheat pixels to the union of the number of predicted winter wheat pixels and the true number, and III represents the ratio of the number of 'missing' winter wheat pixels to the union of the number of predicted winter wheat pixels and the true number. From Figure 8, it can be seen that, except for the seeding-tillering period, for which the value of I is only 79.7%, I is above 83%. The highest value (89.1%) corresponds to the reviving period, indicating that the difference between the area identified as winter wheat in this period and in the jointing-heading period is the smallest. In addition, it can be seen that, for the overwintering period, the value of II is higher than that of III, which indicates that the commission error is more serious than the omission error and that the area identified as winter wheat is larger than during the jointing-heading period.

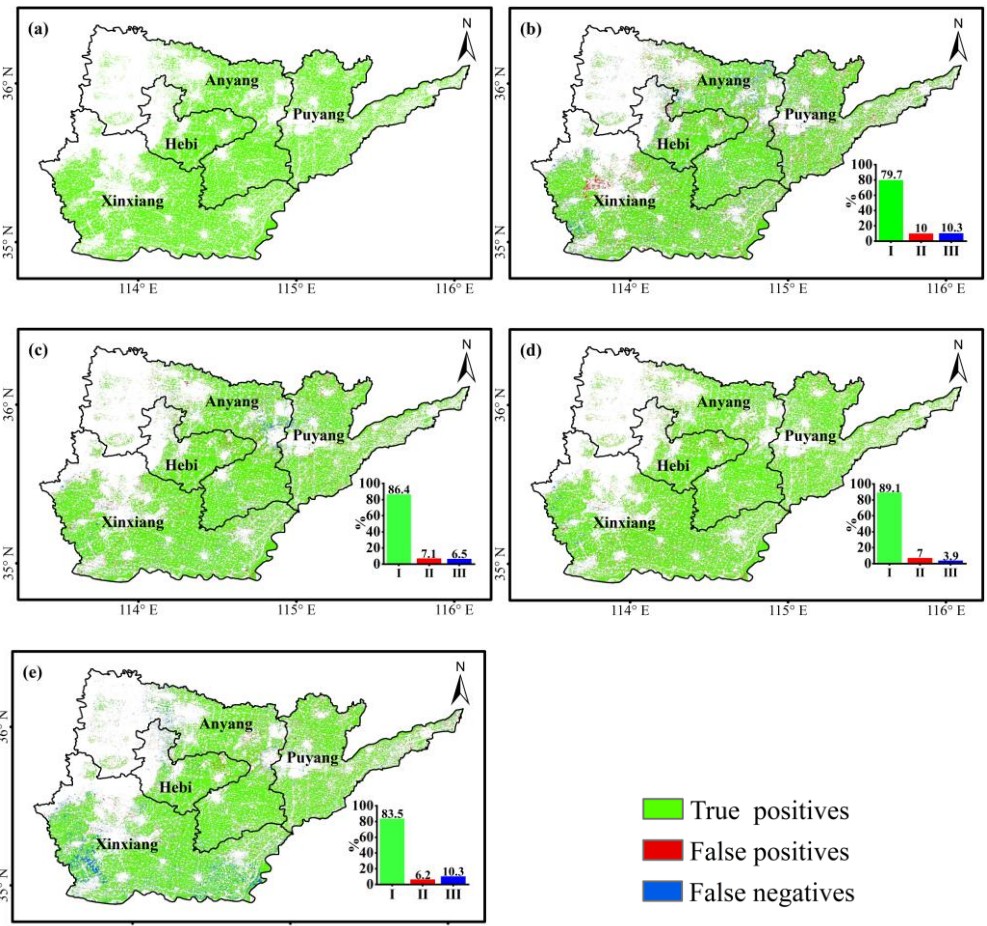

**Figure 8.** Mapping of winter wheat based on deep learning classification. (**a**) U-Net classification map of winter wheat for the jointing-heading period (reference image); it was subtracted from the winter wheat classification map corresponding to other periods to generate difference maps (**b–e**). In the bar charts, I, II and III represents the ratio of the number of correctly predicted, wrongly predicted and 'missing' winter wheat pixels to the union of the number of predicted winter wheat pixels and the true number, respectively.

### 4.3. Extraction Results and Analysis of Winter Wheat Area

We found that the results of the experiments based on the two methods were consistent, and a high level of accuracy was achieved for the identification in both cases. It was also shown that the jointing-heading period is the best period for identifying winter wheat in the study area. Therefore, the area planted with winter area and the extraction accuracy obtained using the two methods were calculated based on the results for this period. These results are shown in Table 5. From the table, it can be seen that the area of winter wheat extracted using the random forest method was 979.67 thousand hectares and that extracted

using the deep learning method was 895.84 thousand hectares. From a comparison with the area of 1012.91 thousand hectares given in the municipal statistical yearbook of the study area for 2020, the accuracy of the extraction for these two methods was 96.72% and 88.44%, respectively. It can be seen that the random forest and the deep learning model both performed well. The calculation of the winter wheat planting area was carried out in GEE in both cases.

**Table 5.** Area of winter wheat extracted and extraction accuracy.

| Classification Method | Area Extracted (Thousands of Hectares) | Extraction Accuracy (%) |
|---|---|---|
| Random forest | 979.67 | 96.72 |
| Deep learning | 895.84 | 88.44 |

## 5. Discussion

### 5.1. The Superiority of Classification Methods

Few previous studies have attempted to explore the accuracy of winter wheat identification at different growth periods, and they all used only one method [24,25]. In this study, two different classification methods—random forest and deep learning—were used for finding the best period on which to base the identification of winter wheat. The random forest method is widely used to identify and extract of crops because of its high classification accuracy, fast training speed, resistance to interference and overall good performance [57–59]. Compared with previous studies, terrain and texture features were added to the random forest classification, which improved the accuracy of the winter wheat identification for the jointing-heading period—the overall accuracy and Kappa coefficient increased by 1.71% and 0.02, respectively. The deep learning method was also used in this study to identify and extract winter wheat in different growth periods. Deep learning has achieved remarkable success in the processing of big data and has been widely used in crop classification and extraction [60,61]; this includes remarkable results in winter wheat mapping. For example, in one previous study, a deep neural network model was established by taking county statistics as the training target for the supervised classification and generating a winter wheat map based on a MODIS image with a resolution of 250 m [62]. This new classification method demonstrates the possibility of carrying out land-use mapping with statistical data as the reference data [62]. U-net network model is a training model that supports a small amount of data, and it can obtain high segmentation accuracy. The trained U-net model can be used to quickly segment the image, automatically learn the required features, perform end-to-end learning and avoid the effects due to artificial feature selection, incomplete features and insufficient representativeness that occur when using the random forest method.

In addition, in order to prove the reliability and applicability of the two methods used in this study. Other methods, such as regression tree [30,63], support vector machine [64,65] classifier, FastFCN [66], DeeplabV3+ [67] semantic segmentation network in deep learning, were also used to evaluate the accuracy of winter wheat identification at jointing-heading period, and the results are shown in Tables 6 and 7.

**Table 6.** The accuracy evaluation indicators of different classifiers.

| Classifiers | User Accuracy | Producer Accuracy | Overall Accuracy | Kappa Coefficient |
|---|---|---|---|---|
| Random forest | **0.97** | **0.95** | **0.97** | **0.96** |
| Cart | 0.94 | 0.94 | 0.93 | 0.92 |
| SVM | **0.97** | **0.95** | 0.94 | 0.92 |

**Table 7.** The accuracy evaluation indicators of different three kinds of networks.

| Networks | Precision | Recall | F1-Score | Accuracy | IoU |
|---|---|---|---|---|---|
| U-Net | **0.94** | **0.93** | **0.94** | **0.94** | **0.88** |
| FastFCN | 0.91 | 0.91 | 0.91 | 0.92 | 0.86 |
| DeeplabV3+ | 0.92 | 0.92 | 0.92 | 0.93 | **0.88** |

The results showed that the random forest classifier had the highest accuracy compared with other classifiers, and the performance of the U-Net network in winter wheat semantic segmentation under a small sample data was better than other networks.

### 5.2. The Key Growth Period for Winter Wheat Identification by Remote Sensing Images

Both experimental results prove the potential of identifying winter wheat by remote sensing images of the key growth periods. Considering that the monitoring of the winter wheat area is of great significance to ensuring food security, it is very important to identify winter wheat and obtain reliable classification results as early as possible. In this study, winter wheat can be accurately identified and mapped its planted area about two months before the harvest, which is beneficial for agricultural departments to monitor the growth of winter wheat and predict the total output of winter wheat [68], and also provide sufficient time for the government to evaluate food security and make reasonable policies. Although several previous studies have identified and classified crops in the early season, their methods are based on the time series curves over the whole growth period to map the planting area of crops [69,70]. In this study, the identification and map of winter wheat were accurately obtained by remote sensing images from the jointing and heading stage, which is not only with non-whole growth period imageries but also provided valuable information about two months before the harvest for monitoring and managing the growth of winter wheat [71,72].

### 5.3. Uncertainty and Outlook

In the above, we have discussed and analyzed the identification of winter wheat from remote sensing data acquired during five different growth periods and using the two methods described. The results that were achieved were generally good. However, there were some limitations to the study, and there are still some problems that need to be further explored.

(1) Sentinel-2 imagery with a spatial resolution of 10 m was used. This may have resulted in there being a variety of land-cover types within individual pixels and, consequently, classification errors. In addition, the western part of the study area contains complex terrain and a large number of mountains, and the land-use patterns near the boundary between Anyang City and Puyang City are highly fragmented. These factors will also influence the classification results. The spatial resolution of the imagery may also have caused some non-winter wheat components to be mixed with the winter wheat samples during the construction of the deep-learning labels, again affecting the classification accuracy and causing classification errors. In follow-up research, remote sensing images with a higher spatial resolution could be used to produce the training, validation and test data sets.

(2) Differences in the quality of the remote sensing images acquired during the different growth periods may also have had some influence on the results. When satellites acquire ground images, they are influenced by both internal and external factors, with the meteorological conditions having the greatest influence. Although cloud removal was carried out, this will still have introduced some errors in the identification of winter wheat.

(3) SAR data were not used in this study. In many studies, it has been found that SAR data can be used as an auxiliary data source to improve the accuracy of the classification based on optical remote sensing data [30,57]. In addition, SAR data are not affected by atmospheric factors such as cloud cover and smoke pollution, which can affect optical remote sensing data, and, thus, can be used to help with the accurate mapping of winter

wheat [73]. Another problem concerns pixel-based classification, which inevitably leads to the 'salt and pepper phenomenon'; this problem can be addressed by adopting an object-oriented classification method [23,74].

(4) The deep learning model contained a large number of hyperparameters, and the best settings of these parameters require further exploration. The number of samples is also an important factor that affected the accuracy of the experiments that were carried out. Whether increasing the size of training and validation data sets could further improve the accuracy remains to be studied. In addition, the two models established in this paper applied to just one particular region, and whether the models are applicable to other regions needs further study.

## 6. Conclusions

This paper is mainly concerned with the accuracy of the identification of winter wheat based on remote sensing data in different growth periods. Winter wheat in part of northern Henan was taken as the research object. Based on the GEE platform and using Sentinel-2 images as the main data source, the separability between winter wheat and other cover types in different growth periods was calculated using the J-M distance method, and a number of vegetation indices, moisture indices, building indices, spectral features, texture features and terrain features were constructed. In combination with field data, winter wheat in different growth periods was identified and extracted using the random forest method. In addition, in order to eliminate the impact of incomplete feature selection, which affects the random forest method, a U-Net semantic segmentation model was used to verify the random forest classification results. The following conclusions were then drawn.

(1) The overall accuracy of the random forest classification was above 91% and the kappa coefficient was above 0.89. The accuracy of the winter wheat identification for the jointing-heading period was the highest—the overall accuracy was 96.90% and the kappa coefficient 0.96. The seeding-tillering period had the lowest accuracy, with an overall accuracy and kappa coefficient of 91.99% and 0.89, respectively. Using the U-Net semantic segmentation model, the values of the IoU for the seeding-tillering, overwintering, reviving, jointing-heading and flowering-maturing periods were 0.78, 0.84, 0.86, 0.88 and 0.82, respectively; the jointing-heading period had the highest value. The precision, recall, F1 score and accuracy for the jointing-heading period were 0.94, 0.93, 0.94 and 0.94, respectively.

(2) The areas of winter wheat extracted using the random forest and deep learning models for the jointing-heading period was 979.67 thousand hectares and 895.84 thousand hectares, respectively; the extraction accuracy was 96.72% and 88.44%, respectively.

(3) The terrain greatly affected the accuracy achieved using the random forest method. The identification of winter wheat was better in areas of flat terrain than in areas of complex terrain; however, the effect on the accuracy of the deep learning classification was unclear.

Using both methods, it was shown that the jointing-heading period was the best for the identification of winter wheat using remote sensing data; the accuracy of the results was also high. These results are of great significance to the quick and accurate estimation of winter wheat planting areas, analysis of the growth of winter wheat and estimates of yields, and the development of precision agriculture and ensuring food security. Future research can focus on further investigations of the accuracy of the identification of winter wheat based on different types of remote sensing data in different regions and using other classification methods. This will help to verify the applicability of the results derived in this study.

**Author Contributions:** Conceptualization, D.P. and S.L.; methodology, D.P., S.L., J.H. and Z.L.; software, D.P., S.L., S.Z., Y.P., J.H. and Z.L.; validation, B.Z., L.Y., J.H., Z.L. and Y.C.; formal analysis, B.Z., D.P. and S.L.; investigation, D.P. and Z.C.; resources, D.P. and Z.C.; data curation, D.P., S.L., Z.C., Y.P., S.Z., Y.C. and S.Y.; writing—original draft preparation, S.L. and D.P.; writing—review and editing, D.P., J.C. and S.L.; visualization, D.P., S.L., S.Z and Y.P.; supervision, D.P.; project administration,

B.Z., D.P. and L.Y.; funding acquisition, B.Z., D.P. and L.Y. All authors have read and agreed to the published version of the manuscript.

**Funding:** This work was supported by the National Natural Science Foundation of China under Grant 42030111 and 42071329. Tsinghua University Initiative Scientific Research Program: (2021Z11GHX002) and the National Key Scientific and Technological Infrastructure project "Earth System Science Numerical Simulator Facility" (EarthLab).

**Conflicts of Interest:** The authors declare no conflict of interest.

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
