# Peer review of "The Accuracy of Winter Wheat Identification at Different Growth Stages Using Remote Sensing"

_remotesensing, doi:10.3390/rs14040893_

Round 1

Reviewer 1 Report

This study used commonly used random forest and deep learning u-net digital image processing techniques and Sentinel 2 data to determine the optimal phenologic stage to obtain measurements of the total area planted of winter wheat using remotely sensed imagery. The best separability was achieved during the joint heading period, and accuracies reached upwards of 96%. Overall, the study is very clear and the importance of such a study is adequately described. Methods are described well, and the results are adequately analyzed. Other than some minor edits and perhaps some minor rearranging of sections, I believe the manuscript should be published.

Lines 18-21: a bit of a run-on sentence; not clear sentence structure.

Line 26: Remove “And”

Line 104: clouds, plural

Line 168: 10 October 10- please remove the other 10

Lines 175-176: Should just be SRTM, not SRTMD

Table 2 is a great addition to the manuscript

Line 260: is GLCM defined before this?

Lines 291-294: awkward phrasing

Line 297: were, not was

Lines 377 and 380: perhaps put the values in parentheses so readers aren’t confused by the (-) sign.

Lines 434-452: this is a lot to have in a figure description. Could you extract much of this to place inn text?

Lines 568-583: Same for this; very long figure description.

Lines 624-659: Perhaps this should be in a section titled “Discussion”.

Line 660: Section 4 is also results; rename this section. Conclusion?

Reviewer 2 Report

The authors carried out an analysis of the accuracy of the identification of winter wheat using remote sensing.

The manuscript is well written, the methodology adopted is adequate for the aim of the research, the results are consistent with the methods used.

However, some changes to the text are required to improve the overall quality of the manuscript.

The introduction is too long, it can be summarized. For instance, the aim of the research, from line 135 to line 151, contains some information described in the methodology.

In paragraph 2.1, Study area, in order to contextualize the research, a better description of the climate of the area is required, in graphic or tabular form.

The authors imposed some conditions on the model but they did not specify the reason.

For example, “The sample points were then randomly divided into two parts: 70% were used for training and classification and 30% for accuracy evaluation, as shown in Table 2” (lines 196-198).

Once again “The number of decision trees was set to 200; all the other parameters were set to their default values” (lines 278-279).

Conclusion is missing in the text.

There are some typos in the text and in the figures caption.

Reviewer 3 Report

The authors evaluated the potential of Sentinel-2 and random forest for monitoring winter crop growth. In a general way, the manuscript is interesting and easy to read. I list here below some comments to improve the presentation.

1. Introduction
I agree with the Jeffries-Matusita distance is a good indicator for evaluating statistical separability. However, some indicators have been proposed for this purpose. You need to clarify the reasons why you chose the Jeffries-Matusita distance.

Could you clarify why you only used Random Forests? You pointed out deep learnings have been used for identifing crops.

Although U-Net is one of the common methods, there are a lot of segmentation methods. More literature reviews were required.

2.2.1 Remote sensing and terrain data
How did you handle the issues that Sentinel 2 MSI data comes in different spatial resolutions?

2.2.3 Sample data
You divided the dataset into two parts: a training set (70%) and a test set (30%). Why did you choose this proportion?

3.1.4 Random forest algorithm and accuracy evaluation index
Which software did you use for generating classifiers based on RF?

4. Results and analysis
You calculated some vegetation indices. Which was effective for identification?

The discussion section is very weak and needs to be improved. There is no comparison of the results with previous results.

Round 2

Reviewer 3 Report

I have no comment.

Author Response

Appreciate your affirmations. We have polished our language.